# Perception of Portuguese Consumers Regarding Food Labeling

**DOI:** 10.3390/nu14142944

**Published:** 2022-07-19

**Authors:** Bruna Silva, João P. M. Lima, Ana Lúcia Baltazar, Ezequiel Pinto, Sónia Fialho

**Affiliations:** 1Scientific-Pedagogical Unit of Dietetics and Nutrition, Coimbra Health School, Polytechnic Institute of Coimbra, 3045-093 Coimbra, Portugal; ana.santos@estescoimbra.pt (A.L.B.); sonia.fialho78@gmail.com (S.F.); 2Center for Innovative Care and Health Technology (ciTechCare), 2411-901 Leiria, Portugal; 3Sustainable Agrifood Production Research Centre (GreenUPorto), 4485-646 Fornelo e Vairão, Portugal; 4The Health Sciences Research Unit: Nursing (UICISA), 3000-232 Coimbra, Portugal; 5Centre for Health Studies and Development, University of Algarve, 8005-139 Faro, Portugal; epinto@ualg.pt; 6Center for Research and Development in Agri-Food Systems and Sustainability (CISAS), Polytechnic Institute of Viana do Castelo, 4900-347 Viana do Castelo, Portugal

**Keywords:** food labels, consumer perception, food and nutrition literacy, consumer choice

## Abstract

Food labels are the first information tool used by consumers in the purchase and consumption of food products. Food labeling is a tool that can influence the consumers’ perception of quality and, in turn, their food choice. This study characterizes and demonstrates the importance of labeling and the degree of consumers’ perception and literacy about food labels through the application of an online questionnaire. The results obtained, in a sample of *n* = 467, showed that Portuguese consumers have the habit of reading the label and recognizing its importance but do not understand all the information contained in the label. They have an easier time understanding the front-of-pack labeling systems, especially those presented through symbols/colors. Thus, it demonstrates the need for greater education and literacy in the areas of food and nutrition so that through the reading and interpretation of labels, consumers can make informed food choices.

## 1. Introduction

Food labeling provides information to the consumer about food products. This information can be varied, however, in Europe, the regulation 1169/2011 made it mandatory to provide information on the label, such as nutrition declaration, name of the food, list of ingredients, use-by date, conditions of use and storage, and origin. In addition to the mandatory information, there may be additional information, such as nutrition and health claims. In addition to the mandatory information, additional information such as nutrition and health claims may be included [1]. By transmitting information essential to consumer choice, labeling plays an important role, allowing consumers to make their choices, thus controlling their health and satisfying their interests [2,3]. Thus, in addition to its informative function, it also functions as a marketing tool and can influence the perception of food quality and, in turn, consumer choice [3]. Moreover, for this reason, research in labeling and the evaluation of its effects on consumers has grown [4]. However, despite its importance, some labels can create false perceptions about the healthiness of products and generate doubts for consumers, so it is important to understand the perception of Portuguese consumers of the labeling of food [5].

The improvement of living conditions and the increase and diversification of food availability associated with globalization and industrialization has been changing the habits and eating patterns of the population [6]. The concept of healthy eating in Europe is increasingly understood, and the practice of healthy eating is perceived as beneficial in contributing to the maintenance/improvement of health. However, there are differences between nutritional recommendations and what consumers eat [2].

Several factors influence the behavior during the act of purchase, and it is essential to verify what they are and how it is possible to intervene so that consumers can make healthier choices [2]. Currently, food consumption is particularly affected by three major trends: Health concerns, environmental sustainability, and convenience/practicality [7,8]. These factors depend on the food product categories (F&V, meat, organic products, among others) [9,10,11]. These trends are particularly important in industrialized countries, where interest in information about the components and production methods of food products has increased significantly [12].

In parallel and in a complementary way, there is a trend of greater consumer concern with the heavy use of pesticides in conventional and intensive agricultural practices, the addition of artificial ingredients, additives, and/or colorings, and the adoption of controversial food technologies, such as genetically modified organisms [8]. This trend, although recent and still without a legally regulated definition, has generated a concept called “clean label” and has contributed to the updating of the way the industry communicates to consumers the form of production and food constituents [8,13,14].

Each individual looks for the answer to their dietary needs and preferences on the label. Thus, sociodemographic factors, such as age, gender, family income, household characteristics, and education level, are some of the factors that can influence the use of food labeling and, consequently, the purchase decision [15].

Most European countries require nutrition information to be presented on food labels [16]. However, consumers have reported difficulty understanding this nutrition information or rarely using it [15]. In addition to nutrition information, claims to reduce nutrition, health, and disease risk may appear on labels [17]. These claims, used by producers to highlight attributes of the food, can influence consumers, as they are used to highlight specific characteristics of the food but do not alert consumers to the content of the remaining nutrients present in the product [17]. Consequently, several countries are introducing front-of-pack labeling (FOP) to discourage the selection of foods with lower nutritional quality and to encourage product reformulation with the manufacturer. Among the different types of FOP labeling systems that have been introduced worldwide, summary indicator systems and nutrient-specific systems are the most commonly implemented, and it is important to determine which system is best understood by the consumer under study and the influence of claims on labeling [17,18,19].

Thus, it is essential that food labeling becomes an accurate and easily interpretable vehicle for information. The main objective of this study was to assess consumers’ perception and literacy regarding food labeling.

## 2. Materials and Methods

A cross-sectional descriptive study was conducted with a non-probability sample of 467 participants. Data were collected between November and December 2021. Participation in the study was voluntary and obtained through informed consent, ensuring confidentiality. The Ethics Committee approved the study of the Polytechnic Institute of Coimbra, technical report n° 124_CEIPC/2021. 

Data were collected using the CAWI methodology—Computer Assisted Web Interview, applying an online questionnaire shared via email and social networks to assess consumers’ perceptions of food labeling.

The questionnaire, written in Portuguese, was organized into three sections and 34 questions (two closed-ended Likert-type scale questions of agreement, twenty-seven multiple-choice questions, one group of true-false questions, and four open-ended questions). It was developed after reviewing the relevant literature used in previous studies related to general food labeling [20,21]. 

The three sections include: (1) Section A—Portuguese Consumers and Knowledge on Nutrition and Healthy Eating, (2) Section B—Food Labeling, (3) Section C—New Concepts in Food Labeling. Eligibility criteria included being 18 years or older, living in Portugal, and being responsible for household purchases. The article uses data from twenty-two of the thirty-four questions applied. A section of the questionnaire was developed through the Nutrition Knowledge Questionnaire, developed by Turrell and Kavanagh, to assess the knowledge on nutrition and healthy eating [20].

Data were analyzed using the Statistical Package for Social Science—SPSS, version 22. A critical significance level of 5% was considered. A descriptive statistical analysis of the data was performed and obtained means and standard deviations. Differences between groups were assessed by the contingency tables and the nonparametric test of independence chi-square.

## 3. Results

### 3.1. Sociodemographic Characteristics of the Study Sample

From the total sample (*n =* 467) of the adult population living in Portugal who participated in the study, 82.4% of the responses were from female participants, with the remaining 17.6% of the results corresponding to males (Table 1).

The predominant age group in the study was “18–25 years old”, with a representation of 33.6%.

As for the educational qualifications of the study participants, it was observed that most respondents had a higher education degree (68%) (Table 1).

### 3.2. Anthropometric Characterization

The mean weight of the sample under study corresponds to 66.77 ± 14.38 and the mean height to 165.4 ± 8.15. Analyzing the Body Mass Index (BMI) of the sample, it was observed that more than half of the study sample (63.4%) is in the “normal weight” category and 33.8% of the sample fall into the pre-obesity and/or obesity category (Table 2).

About 72% of individuals reported not having any diagnosis of pathologies.

### 3.3. Knowledge about Food & Nutrition

The average score obtained in the questionnaire on knowledge of food and nutrition was 16.49 ± 3.065, on a scale of values from 0 to 20 (Table 3).

When analyzing the respondents’ knowledge in the study about the recommended daily value of salt, only 34.5% of the sample was able to correctly identify the value. Regarding the identification of the recommended daily value for sugar consumption, 46.5% of the sample correctly identified the recommended daily amount.

There was a positive correlation between the variable “age group” and the variable “Knowledge in Food and Nutrition” (r = 0.140; *p* = 0.016). The higher the age of the consumers, the higher the level of knowledge (Table 3).

There was also a positive correlation between the “Knowledge in Food and Nutrition” and the “Literary Qualifications” (r = 0.003; *p* = 0.013), with consumers who have a higher level of education, demonstrating a higher level of knowledge (Table 3).

### 3.4. Food Label Behavior

According to Table 4, 84.2% of the study sample read the food label. Analyzing the label reading frequency, it was found that only 46.7% have the regular habit of reading it (Table 4).

According to the consumer, the main reason identified for reading the label is practicing a healthy diet.

It was found that female individuals read the information on labels the most (*p* = 0.018). It was also found that men read food labels out of curiosity and beliefs about them, while women identified the goal of practicing healthy eating and the prevention and verifying of information on food allergies and/or intolerances as the main reasons for reading (*p* = 0.008) (Figure 1). 

Additionally, it is found that there is an association between the BMI classes and reading the label for adopting a healthier diet (*p* = 0.019). The “Healthy Weight” BMI class has a greater relationship with reading the label (Table 4).

### 3.5. Food Label Perception

Only 20.6% of the sample understands all the information on the label (Table 5).

The main reason identified for the difficulty in understanding the information contained on the labels is the lack of knowledge of technical and scientific terms (Figure 2).

When analyzing the results, it was found that there is a positive association (*p* = 0.019) between the understanding of the label information and the age group of the participants. The age group between 40–49 years and 50–64 years has a superior level of understanding (Figure 2).

### 3.6. Importance of Labels and Purchase Determinants

According to Figure 3, of all the information present on food labeling, the information that the sample highlighted as the most important was: The expiration date, nutritional information, and the ingredient list.

Of the determinants under study for the purchase of food products, the one that showed the greatest impact was “health and nutrition”, followed by the determinant of family preferences (Figure 4).

There is a significant positive association between the “BMI class” and the purchase determinants, namely the “Health and Nutrition” determinant and the obesity class (*p* = 0.011).

### 3.7. Perceptions and Behaviors on Nutrition Label

Considering that nutrition labeling is mandatory on food product labels, it was assessed that the consumer considers this obligation very important (Table 6).

The main reason for reading the nutritional label observed is the acquisition of the food product for the first time. Consequently, it was found that the time spent by the consumer analyzing the label varies mainly between 30 s and 1 min (Table 6).

When asked if nutritional information would impact the purchase of food products, 82.4% of the sample reported positively. The information that influences the food choices described is mainly: The nutritional table, the caloric value, the amount of fat, the number of carbohydrates, and the list of ingredients (Table 6).

Regarding nutritional information, the most sought after by consumers are sugar, carbohydrates, salt, and lipids contents.

### 3.8. Impact of the Presentation of Nutritional Information

Evaluating the preference of the Portuguese consumer regarding the form of presentation of nutrients, it was observed that there is no consensus since 25.7% prefer the information per serving (30 g), 30.4% per unit of consumption, and 32.1% prefer the presentation of information by 100 g of product (32.1%). It was observed that there is no dependence between the preferences of presentation of the portion size and the age group of the participants. Of the FOP systems under study, the nutritional traffic light was the format preferred by the consumer (Table 7). It was also observed that most consumers reported understanding the information transmitted in the FOP system under analysis (Figure 5).

Of the study sample, 84.6% reported that they would change their eating behavior by decreasing the intake of products classified as nutritionally unbalanced according to the FOP system (Figure 5). 

### 3.9. Impact of Nutritional and Health Claims

It was found that 45% of the sample identified that the statements on the front of packages help to decide which products to buy. Practically, half of the sample reported that this type of information helps in decision-making when they are under pressure, ending up purchasing products with this type of information on the label. In terms of understanding the allegations, about half of the sample (48.6%) claims to understand their meaning.

### 3.10. New Food Label Concepts

It was observed that, on average, the product preferred by the consumer should be: “Free from genetically modified organisms” (3.63 ± 1.10), “free from artificial” (3.89 ± 0.99), “natural” (4.16 ± 0.78), “local” (4.07 ± 0.845), “minimally processed” (3.86 ± 1.05), “additive-free” (3.96 ± 0.95), and “free from colorants and preservatives” (3.90 ± 0.95) (Figure 6).

### 3.11. Food Packaging

It can be seen in Figure 7, that consumers, on average, agree that the packaging should consist of a “simple and short list of ingredients” (3.95 ± 1.01). It was found that “the company’s brand” must be present and visible (3.76 ± 0.89), as well as “information on environmental sustainability” (3.79 ± 0.89). Regarding the “origin of the food”, most consumers (4.31 ± 0.76) fully agree with the presence of this information on the packaging. Consumers who read package labels consider “simplicity of the ingredient list” and “company brand” as the main factors to consider in food packaging. 

### 3.12. Clean Label Trend

It was found that, with regards to the Clean Label trend, consumers still do not have a formed opinion, as seen in Table 8. This is true since, on average, most respondents neither agreed nor disagreed with the assumption that clean label products are “more nutritious” (3.1 ± 0.64), “safer” (3.19 ± 0.64), or “healthier” (3.24 ± 0.70).

## 4. Discussion

The main objective of this study was to examine Portuguese consumers’ attitudes towards the labeling of food products. A questionnaire composed of several questions was applied to gather different information about the participants’ perceptions on this topic. 

As previously observed in other studies, it was found that female consumers consult and use food labels more as a group when compared to male consumers [21,22,23,24,25,26].

Females show a greater concern when buying food and take longer to make their choices, reporting that labels influence their food choices. Consumer literacy (knowledge and understanding of nutritional information) is another factor that strongly influences food choices [27,28,29].

Analyzing the impact factor that determines and influences the moment of purchase of food by consumers, “health and nutrition” was identified as a dominant factor, and when analyzing the reason for reading the labels, the factor frequently reported by the sample under study was the intention of practicing a healthy diet. The recognition of the importance of healthy eating for health, the nutritional value of foods, and nutritional knowledge are factors positively associated with the frequency of use of food labels, witnessing an association between the use of labels and the objective of adopting healthy practices, as observed in previous studies [25,30,31,32].

Food labeling plays an important role in the transmission of food and nutrition information, being considered and interpreted by the consumer as a reliable source of information to be used in the selection of food products [25]. However, despite the importance attributed to labeling, it was observed that only 20% of the sample under study understands all the information present on the label, and it is possible to perceive that there are challenges in terms of consumer understanding and proper use of the information present on the label. The self-reported reasons that most affect the understanding of the label information were the reduced text font size, excess of information, lack of knowledge, and confusion regarding technical terms.

In addition to the importance of food labeling, the consumer attaches importance to mandatory nutritional labeling. Thus, it is important to clarify which factors the consumer should analyze when choosing a product. In this study, it was observed that the nutritional information most sought after by consumers was sugar content, carbohydrates, salt, saturated fatty acids, and lipids. However, despite these being the nutritional information most sought after by consumers, it was possible to analyze that more than half of the sample could not identify the recommended consumption values of salt and sugar recommended by the WHO. This fact demonstrates that the consumer is unaware of the salt and sugar content of commonly consumed food since the consumption of sugar and salt in Portugal is far above the limit recommended by the WHO, as observed in another study [21]. 

Thus, although consumers report the use and reading of labels, it was not evidenced that they understand and interpret the information most appropriately, as previously observed by other authors [25,33,34].

Considering the consumers’ difficulties in interpreting the nutritional information and labeling, the simplification formats for the presentation of nutritional information that would be chosen by the sample under study were evaluated. More than half of consumers preferred presenting this information on the front of the package through the “nutritional traffic light” system. In addition to this preference, most consumers demonstrate that they understand the information in this simplified system. When asked whether consumers would reduce their intake of a particular food if it did not present optimal nutritional characteristics, it was found that 84.6% of consumers would reduce their consumption of that product. Some studies suggest that FOP labels improve consumers’ ability to distinguish between healthier and less healthy foods. However, there is still no consensus on this topic. The FOP labels appear to help consumers recognize which foods are healthier. However, there is still little evidence that this knowledge significantly impacts actual shopping behavior. As for the impact of multicolor system labels, some studies point to positive results in purchasing healthier foods [35,36]. 

Nutrition labeling became mandatory in the EU in 2011, going into effect in December 2016. This mandate required nutrition information to be declared per 100 g/mL to allow the comparison of foods of different sizes. According to previous studies, it is known that the variation of the declared portion size can affect consumers’ understanding of nutritional information [37]. However, the declaration of nutrition information regarding serving size and the total number of servings is also encouraged, although there are no pre-set values [16,37]. Thus, analyzing the consumer’s preference for the presentation of nutritional information, it was verified which presentation format was preferred by the consumer. There was no consensus in the answers presented: 100 g, consumption unit, or portion. However, the small majority (32.1%) preferred the presentation of information per 100 g.

Clear and easy to understand for the consumer. According to the results, the variation in the format of presentation of nutritional information and the sizes of the portions presented can make it difficult for the consumer to understand the information contained on the label and can compromise food choices. There is no absolute consensus on which information presentation format is preferred. It is also apparent in other studies that portion size is one of the items that is least understood on food labels by consumers [38,39]. Although not statistically significant, the fact that the small majority indicated that they prefer the presentation of nutritional information per 100 g, may be since it is the information that allows the comparison of nutrients with other foods in a more simplified way, however, further studies in the Portuguese population will be needed on the impact of the presentation of nutritional information on food choices and on the understanding of the information, as well as on the definition of the best format to present the portion size on food labels, to provide nutritional information.

Nutrition and health claims can help consumers make healthier food choices [40]. In the sample under study, it was observed that half of the sample agreed or fully agreed with the help this information offers in the purchase of food products.

According to the results, food with health and nutritional claims helped consumers at the time of the purchase and facilitated their decision-making. This may be shaping consumers’ knowledge regarding the perception of the healthfulness of products, making food with claims generally appear healthier and consequently influence food purchase intentions, moderated by the perception that the products purchased are healthier. It was also found that half of the sample claims to understand the meaning of claims, such as “low sugar content, fiber source”, and these types of claims facilitate decision-making at the time of purchase. These situations have been verified in previous studies [17,40,41].

The emergence of the new trend associated with the healthiness and sustainability of food products, the “clean label” trend, led the authors of this study to analyze the perceptions of Portuguese consumers about the impact of this trend on their beliefs and consumer perceptions. When analyzing the new labeling concepts, such as consumption of organic, local ingredients, free from genetically modified organisms, additives, dyes, and preservatives, it was observed that more than half of the sample believes that the daily consumption of food should be regulated for these characteristics, common to the “clean label” trend. Most of the sample did not agree or disagree with the statements in the study, like “the clean label products are healthier, more nutritious, and safer”.

It is concluded that the consumers under study may not yet have enough information about this tendency to make their conclusions. 

These consumption factors have already been observed in other studies, and it is currently possible to observe a distance from the consumer of highly processed ingredients in the food industry and increasing demand for products that contain familiar, minimally processed ingredients and short lists of ingredients, such as the study sample mentioned [8,42,43].

To sum up, the results highlight some key points concerning knowledge, interpretation, and use of food labels. However, they must be interpreted considering some limitations: The study was based only on self-reported responses, not including measures, and the sample is not statistically representative of the entire Portuguese population.

## 5. Conclusions

After this study, it is noticeable that the Portuguese population understands the importance of labeling food products and that the information contained in the labels has an impact on consumers’ purchasing decisions. However, despite its due importance being recognized and the existence of acceptable label reading rates, it appears that most consumers do not fully understand the information provided through food labeling. The lack of knowledge regarding the concepts and terminology present on food product labels could be impacting consumers’ behavior and dietary pattern without them realizing it, so the information must be exposed in a clear, precise, and evident way.

Regarding the FOP systems, a trend of preference for labels with color systems and/or symbols is identified. From the consumer’s point of view, these seem to facilitate the understanding of the information transmitted, ultimately impacting the purchase of food products. To validate this identified trend, it would be interesting to carry out complementary studies to analyze this system against others, such as the health star rating and health warnings.

In terms of consumer behavior, it is concluded that there is no clear distinction between formats for the presentation of nutritional information, with no significant differences in the choice of the option per portion or 100 g. This fact may be due, once again, to the lack of a clear understanding of these concepts. On the contrary, it is concluded that the consumer’s appetite for healthier, sustainable, natural, organic products are significant. However, we cannot directly associate these characteristics with the new clean label labeling trend.

In short, it is noticeable that in Portugal, as in other European countries, there is a need for greater investment in consumer education and literacy in nutrition and healthy eating. Through this training, consumers can consciously make healthy food choices.

## Figures and Tables

**Figure 1 nutrients-14-02944-f001:**
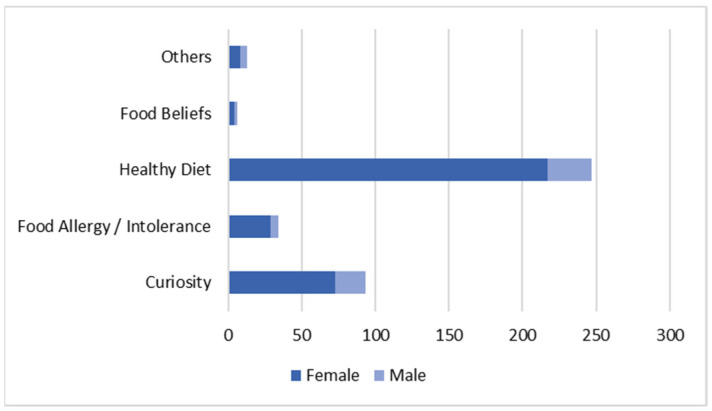
Label reading determinants (*n =* 467).

**Figure 2 nutrients-14-02944-f002:**
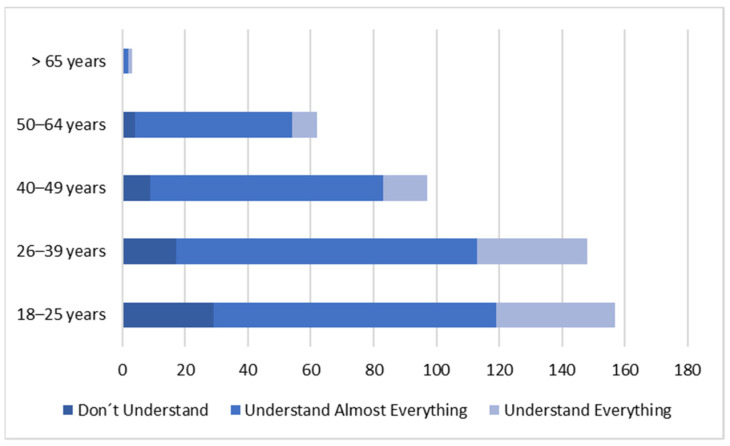
Food Label Perception. N: Number of individuals (*n =* 467).

**Figure 3 nutrients-14-02944-f003:**
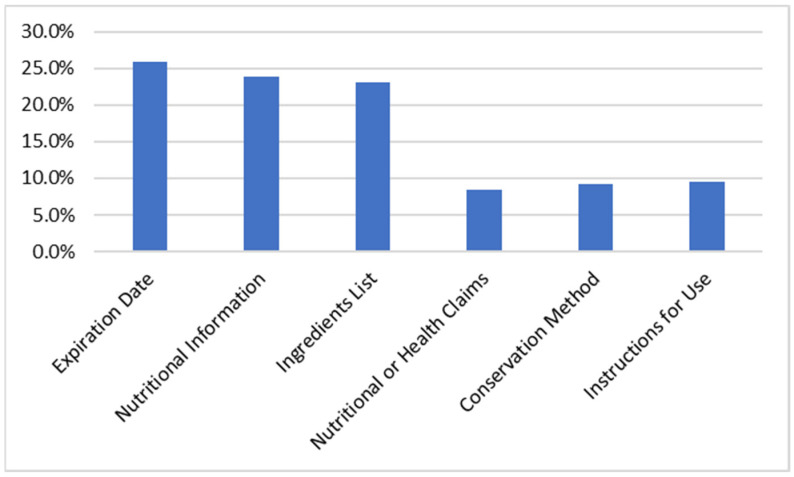
Most important information on the label (*n =* 467).

**Figure 4 nutrients-14-02944-f004:**
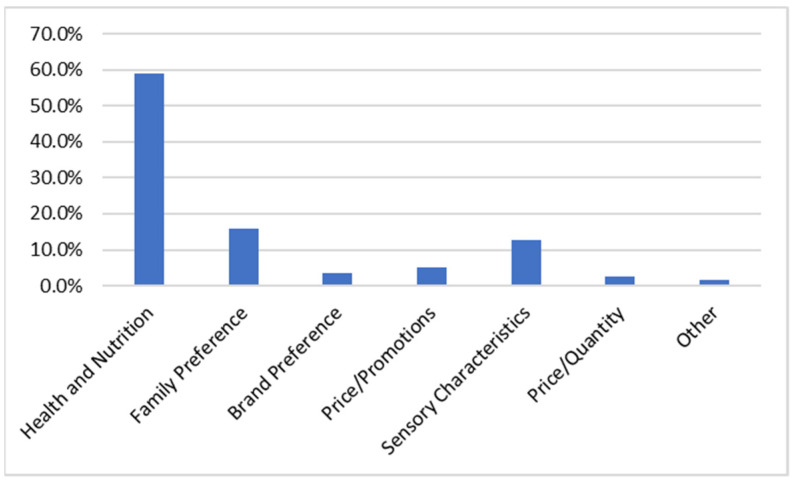
Purchase Determinants (*n =* 467).

**Figure 5 nutrients-14-02944-f005:**
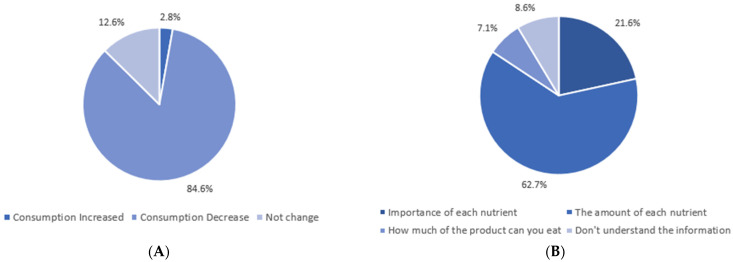
The impact of FOP system on consumer behavior (*n =* 467). (**A**) Impact of the FOP System on food consumption, (**B**) Impact of the FOP System on consumer understanding.

**Figure 6 nutrients-14-02944-f006:**
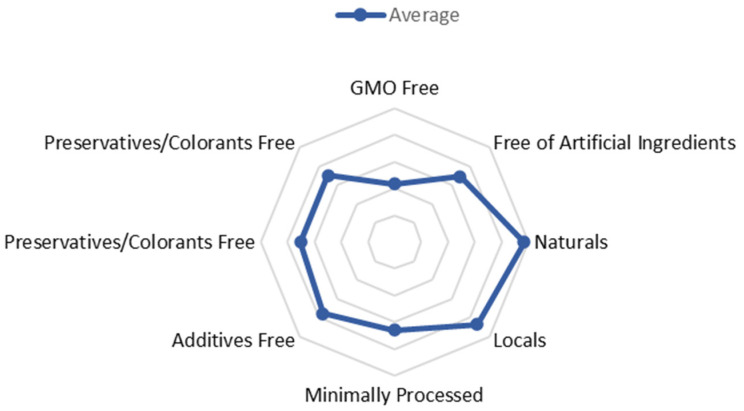
New Concepts in Food Labeling (*n =* 467).

**Figure 7 nutrients-14-02944-f007:**
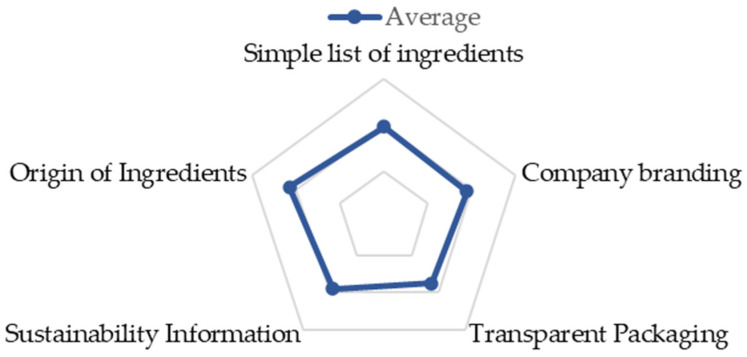
Food Packaging (*n =* 467).

**Table 1 nutrients-14-02944-t001:** Sociodemographic characteristics of consumers (*n* = 467).

**Age Group (Years)**	** *n* **	** *%* **
18–25	157	33.6
26–39	148	31.7
40–49	97	20.8
50–64	62	13.3
>65	3	0.6
**Gender**	** *n* **	** *%* **
Male	82	17.6
Female	385	82.4
**Education Level**	** *n* **	** *%* **
Basic Education	23	4.8
High School	127	27.2
University Education	317	68

**Table 2 nutrients-14-02944-t002:** Anthropometric characterization of consumers (*n* = 467).

	** *n* **	$\bar{X}$
Height	467	165.4
Weight	467	66.77
**IMC**	** *n* **	** *%* **
Underweight	13	2.8
Normal weight	296	63.4
Pre-obesity	108	23.1
Obesity	50	10.7

**Table 3 nutrients-14-02944-t003:** Knowledge about Food and Nutrition (*n* = 467).

**Recommended Daily Value of Salt**	** *n* **	** *%* **
Wrong	306	65.5
Right	161	34.5
**Recommended Daily Value of Sugar**	** *n* **	** *%* **
Wrong	250	53.6
Right	217	46.4
**Knowledge about Food Nutrition** **(KFN-questionnaire)**	** *n* **	$\bar{X}$
	467	16.49

**Table 4 nutrients-14-02944-t004:** Food Labeling Perceptions and Behaviors (*n =* 467).

**Reading Food Labeling**	** *n* **	** *%* **
No	74	15.8
Yes	393	84.2
**Frequency in Label Reading**	** *n* **	** *%* **
Regularly	218	46.7
Occasionally	167	35.8
Rarely	68	14.6
Never	14	3
**Reasons for Reading the Labeling**	** *n* **	** *%* **
Intention to have a healthy diet	247	52.9
Curiosity	93	19.9
Food Allergies or Intolerances	34	7.3

**Table 5 nutrients-14-02944-t005:** Food Labeling Perceptions (*n =* 467).

**Understanding the Information on the Label**	** *n* **	**%**
Don’t understand the information	59	12.6
I understand most of the information	312	66.8
I understand all the information	96	20.6
**Reasons That Affect the Understanding of Information**	** *n* **	**%**
Small Print	117	15.4
Confusion between terms	102	13.5
Lack of knowledge of technical information	159	21
Too much information	109	14.4
Difficulty in interpretation	104	13.7

**Table 6 nutrients-14-02944-t006:** Food Labeling Perceptions and Behaviors (*n =* 467).

**Importance of Nutrition Labeling**	** *n* **	** *%* **
Very Important	364	77.9
Important	96	20.6
Not too Important	6	1.3
**Average time to read Nutrition Labeling**	** *n* **	** *%* **
Less than 30 s	109	23.3
Between 30 s and 1 min	257	55
**Reason for Reading Nutrition Label**	** *n* **	** *%* **
Product purchased for the 1st time	355	81.1
New product on the market	156	35.6
Special dietary needs in the household	144	32.9
**Key Nutritional Information**	** *n* **	** *%* **
Ingredients List	208	18.7
Nutritional Table	227	20.4
Calories	176	15.8
Amount of fat	192	17.2
Amount of carbohydrates	146	13.1
**Nutrition Labeling Interferes with Food Choices**	** *n* **	** *%* **
Yes	385	82.4
No	82	17.6

**Table 7 nutrients-14-02944-t007:** Preferred Labeling System (*n =* 467).

Label 1	Label 2	Label 3
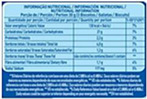	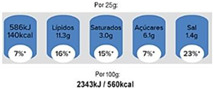	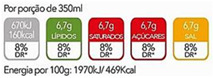
26.6%	4.1%	69.3%

**Table 8 nutrients-14-02944-t008:** Food Packaging (*n =* 467).

Determinants for Food Packaging	*p*-Value
Simple list of ingredients	0.032 ^1^
Company branding	0.003 ^1^
Transparent Packaging	0.537
Sustainability Information	0.386
Origin of Ingredients	0.850

^1^ Differences with statistical significance.

## Data Availability

The work was part of Bruna Silva’s postgraduate.

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
