# Peer review of "Perception of Portuguese Consumers Regarding Food Labeling"

_nutrients, 2022, doi:10.3390/nu14142944_

Round 1
Reviewer 1 Report
The authors explore an interesting topic, I enjoyed read the paper which is well written and organized.
The label of a food product carries a lot of information that influences the consumer's decision to buy it. For the modern consumer, food labelling is a basic source of knowledge about the product and the way it is used. From the food label the consumer learns about the specific properties of the product (e.g. its impact on health, the production methods used, etc.). Therefore, apart from interesting graphics and attractive colours, the label must contain - above all - reliable information about the product so that the consumer can make a conscious choice about the food he consumes.
Reviewer's comments:
1. In Abstract and Materials and Methods, please write on which sample the study was conducted (N=...). This was presented only in chapter 3.1.
2. 80 line: I suggest you add which method was used - CAWI method
3. 87-91 lines: were all three parts of the questionnaire used to develop the article? I propose to write: The article uses data from selected (how many?) questions on....
4. 4. 99 line: Socio- demographic characteristic of the study sample
5. 5. 107 line: Table 1 – characteristic
6. In the titles of tables and diagrams please include the size of the study sample (N=...)
7. 142 line: N=467
8. 239-243 lines: a sentence about the limitations of the study and the article should be at the end of the Discussion
9. What is the significance of the study for other European countries?
Reviewer 2 Report
The submitted paper is easy to read. The topic is relevant but I guess if the conducted study can be rewritten in a less descriptive approach.
The main concerns I Have are:
- the convenience sample;
- when speaking of food labelling is evident that the percetion of consumers may depend on differnt food macrocategories: (perishable foods, baby food, fresh/frozen foods) is nont clear to me if the interviewed have give information on their purchase habits.
I suggest the author to rewrite the results and discussion, if possible, with a less descriptive approach. I guess if you have the possibility to discriminate the sample between interviewed that are the only purchasers in their family and the ones that are in a different situation.
I give the authors a few suggestions on the text.
- LINES 23-33 I suggest the authors to specify that information ofn fodd products can be distinguished in mandatory and additiona
- LINES 44-46: : the perception/importance percieved by consumers of health concerns, environmental sustainability, and convenience/practicality may depend from the food products catergories (F&V, meat, ….). I suggest the authors to insert several references o nthis topic.
- LINES 80-81: the authors should specify the way they subministered the questionnaire on line (using mialing lists, blogs and so on …
- LINE 107: pay attention to the total percentage of Education Level (100,1%)
- TABEL 5 and figure 2 report the same information I suggest to cut figure 2
- LINES 200-201: the figure has a legenda that must be rewritten.
- LINE 211: I think that “organic” can’t be defined as a “New Food Label Concept” … it is an EU certified production/processing process since long time.
- LINES 244.247: the sentences about the female over representation is not clear to me.
Round 2
Reviewer 2 Report
The authors implemented the paper following the reviewers' suggestions. suggest to accept the paper in the present version.